# High-Dosage NMN Promotes Ferroptosis to Suppress Lung Adenocarcinoma Growth through the NAM-Mediated SIRT1–AMPK–ACC Pathway

**DOI:** 10.3390/cancers15092427

**Published:** 2023-04-23

**Authors:** Mingjiong Zhang, Jiahua Cui, Haoyan Chen, Yu Wang, Xingwang Kuai, Sibo Sun, Qi Tang, Feng Zong, Qiaoyu Chen, Jianqing Wu, Shuangshuang Wu

**Affiliations:** 1Jiangsu Provincial Key Laboratory of Geriatrics, Department of Geriatrics, The First Affiliated Hospital with Nanjing Medical University, Nanjing 210029, China; 2Department of Epidemiology, School of Public Health, Nantong University, Nantong 226019, China; 3Department of Pathology, Medical School, Nantong University, Nantong 226001, China; 4NHC Key Laboratory of Antibody Technique, Nanjing Medical University, Nanjing 210029, China; 5Centre for Assisted Reproduction, Shanghai First Maternity and Infant Hospital, Tongji University School of Medicine, Shanghai 200071, China

**Keywords:** NMN, NAM, NAMPT, ferroptosis, lung adenocarcinoma

## Abstract

**Simple Summary:**

Nicotinamide mononucleotide (NMN) is the physiological circulating nicotinamide adenine dinucleotide (NAD) precursor thought to elevate the cellular level of NAD^+^ and ameliorate various age-related diseases. Here, we investigate how high-dose NMN functions in lung adenocarcinoma. We show that excess nicotinamide (NAM) is produced through the metabolism of high-dose NMN, while the overexpression of nicotinamide phosphoribosyltransferase (NAMPT) can significantly decrease intracellular NAM content and, in turn, boost cell proliferation in vitro and in vivo. Mechanistically, high-dose NMN promotes ferroptosis through NAM-mediated SIRT1–AMPK–ACC signaling. This study highlights the tumor influence of NMN at high doses in the manipulation of cancer cell metabolism, providing a new perspective on clinical therapy in patients with lung adenocarcinoma.

**Abstract:**

Background: Nicotinamide mononucleotide (NMN) is the physiological circulating NAD precursor thought to elevate the cellular level of NAD^+^ and to ameliorate various age-related diseases. An inseparable link exists between aging and tumorigenesis, especially involving aberrant energetic metabolism and cell fate regulation in cancer cells. However, few studies have directly investigated the effects of NMN on another major ageing-related disease: tumors. Methods: We conducted a series of cell and mouse models to evaluate the anti-tumor effect of high-dose NMN. Transmission electron microscopy and a Mito-FerroGreen-labeled immunofluorescence assay (Fe^2+^) were utilized to demonstrate ferroptosis. The metabolites of NAM were detected via ELISA. The expression of the proteins involved in the SIRT1–AMPK–ACC signaling were detected using a Western blot assay. Results: The results showed that high-dose NMN inhibits lung adenocarcinoma growth in vitro and in vivo. Excess NAM is produced through the metabolism of high-dose NMN, whereas the overexpression of NAMPT significantly decreases intracellular NAM content, which, in turn, boosts cell proliferation. Mechanistically, high-dose NMN promotes ferroptosis through NAM-mediated SIRT1–AMPK–ACC signaling. Conclusions: This study highlights the tumor influence of NMN at high doses in the manipulation of cancer cell metabolism, providing a new perspective on clinical therapy in patients with lung adenocarcinoma.

## 1. Introduction

Aging is characterized by the functional decline of tissues and organs and the increased risk of aging-associated diseases, such as atherosclerosis, heart disease, stroke, Parkinson’s disease, Alzheimer’s disease, etc. [1,2]. Therefore, anti-aging interventions have been developed to delay aging and the onset of age-associated diseases to extend one’s healthspan and lifespan [3,4]. NMN is a biologically active nucleotide that is formed by the reaction of ribose and nicotinamide, which is an ideal candidate for anti-aging drugs [5,6]. NMN is an intermediate in the nicotinamide adenine dinucleotide (NAD) salvage synthesis reaction, generating NAD under the action of nicotinamide phosphoribosyltransferase (NAMPT) [7]. However, few studies have directly investigated the effects of NMN on another major ageing-related disease: tumors [8]. An inseparable link exists between aging and tumorigenesis, especially involving aberrant energetic metabolism and cell fate regulation in cancer cells [9].

Lung cancer is the second main cause of death in cancer patients worldwide [10]. According to the Global Cancer Epidemiological Survey, the incidence and mortality of lung cancer are positively correlated with age. Compared with younger lung cancer patients, the survival time of lung cancer patients over 60 years old is significantly lower [11]. Aging may be the main factor promoting the occurrence and progression of lung cancer. Senescent endothelial cells lead to a discontinuous endothelial lining and defective basement membrane around the tumor; so, these cells lose their role in limiting tumor growth and metastasis [12]. The inflammation and hyporesponsiveness caused by aging also limit the effectiveness of lung cancer treatment [13,14]. NMN appears to have yet-unconfirmed roles in lung adenocarcinoma and may therefore represent a target for therapeutic strategies.

Ferroptosis is an iron-dependent regulated cell death mechanism regulated by lipid-based accumulation, tightly linked with the oxidative stress response and cysteine metabolism [15]. Accumulating evidence indicates that ferroptosis is involved in the pathophysiologic processes of various diseases, including cancers, and can act as a natural barrier to tumor progression [16,17]. Ferroptosis is characterized by the accumulation of peroxides of phospholipids enriched with polyunsaturated fatty acids and ROS [18]. This feature is inextricably linked to mitochondrial respiration regulated by NAD^+^/NADH. NMN may be involved in the regulation of ferroptosis [19].

In this study, we clarified the effects of high-dose NMN in inhibiting lung cancer growth by inducing non-apoptotic forms of regulated cell death. Furthermore, using multiple death-signal-related inhibitors, we found that the firm tumor suppressive ability of NMN at high concentrations mainly relied on the iron-mediated cell death pathway: ferroptosis. Given the findings that NMN is either converted to NR or NAD, and then to NAM in vivo, we tested the above metabolites and found that NAM overload plays a vital role in suppressing lung adenocarcinoma growth. Mechanistically, high-dose NMN promotes ferroptosis through NAM-mediated SIRT1–AMPK–ACC signaling.

## 2. Materials and Methods

### 2.1. Cell Culture and Reagents

Both the human lung adenocarcinoma A549 (RRID: CVCL_0023) and SPCA1 (RRID: CVCL_6955) cell lines were obtained from Jiangsu Provincial Key Laboratory of Geriatrics (Nanjing, Jiangsu, China), and erastin/RSL3-resistant lung adenocarcinoma cell lines were kindly provided by the Key Laboratory of Antibody Technique of National Health Commission, Nanjing Medical University (Nanjing, Jiangsu, China). The A549 (Shanghai Biowing Biotechnology Application, Shanghai, China) and SPCA1 (KeyGEN BioTECH, Shanghai, China) cell lines were verified via STR analysis. In addition, the cells were tested for the presence of mycoplasma infection using a PCR assay. All the cell lines were cultured in DMEM (Sigma-Aldrich, Darmstadt, Germany) with 100 μg/mL streptomycin, 100 U/mL penicillin (Gibco, Billings, MT, USA), and 10% fetal bovine serum (FBS) (Invitrogen, Waltham, MA, USA) in a humidified atmosphere (Thermo Fisher Scientific, Waltham, MA, USA) that was 5% CO_2_. All cells used for experiments were actively passaged for less than 3 weeks and were in the logarithmic growth phase.

### 2.2. Lentivirus Construction and Infection

To construct NAMPT-overexpressed plasmids, human NAMPT cDNA was synthesized and cloned into pSLenti-EF1-mCherry-F2A-Puro-WPRE2-CMV-MCS vector with OBiO (Shanghai, China), and cells were infected by lentivirus vectors carrying specific target genes in the presence of polybrene (10 μg/mL) and screened with 2.5 μg/mL of puromycin (Sigma-Aldrich) for 2 weeks to obtain stable cell lines. All indicated proteins were identified via Western blotting.

### 2.3. Western Blot Assay

Western blot assay was performed as reported previously [20]. Briefly, the total protein from the cell samples was extracted using RIPA lysis buffer supplemented with protease and phosphatase inhibitors (Beyotime, Shanghai, China). A total of 40 µg of protein was processed for the next analysis. The protein was separated with 10% SDS-PAGE, transferred onto polyvinylidene difluoride (PVDF) membranes (Millipore, Darmstadt, Germany), and blotted for primary antibodies: cleaved-caspase3 (Proteintech, Wuhan, China), PARP (Proteintech), β-actin (Proteintech), p-AMPK (Cell Signaling Technology, Danvers, MA, USA), AMPK (Cell Signaling Technology), p-ACC (Cell Signaling Technology), ACC (Cell Signaling Technology), NAMPT (Proteintech), and SIRT1 (Proteintech), followed by the corresponding horseradish-peroxidase-conjugated secondary antibodies (Proteintech) for 1 h at room temperature. The membranes were processed and visualized using BeyoECL Star kit (Beyotime). β-actin was used as the loading control.

### 2.4. Cell Proliferation Assay

The proliferation of A549 and SPCA1 cells was assessed using a cell counting kit-8 (CCK8) (MCE, Shanghai, China) and a colony formation assay with 0.1% crystal violet staining (Beyotime). For the CCK8 assay, A549 (5000 cells) and SPCA1 (5000 cells) cells were seeded into 96-well plates overnight and treated with the following drugs, according to the manufacturer’s instructions: nicotinamide mononucleotide (NMN) (Uthever, Carpinteria, CA, USA), Z-DEVD-FMK (50 µM, MCE), procaspase-activating compound 1 (PAC-1) (500 nM, MCE), necrostatin-1 (Nec-1) (50 µM, MCE), chloroquine (CQ) (25 µM, MCE), ferrostatin-1 (Fer-1) (10 µM, MCE), E-daporinad (FK866) (5 nM, MCE), CAY10602 (10 µM, MCE), GSK621 (10 µM, MCE), dorsomorphin (10 µM, MCE), or selisistat (15 µM, MCE). We added 10 μL of CCK8 solution to each well. After 1 h, the absorbance at 450 nm was measured using an automatic microplate reader (Thermo Fisher Scientific). For the CCK8 assay, we adjusted the pH using NaHCO_3_ because high-dose NMN-induced acidic environments inhibit cell growth, which interferes with the actual proliferation level of the tumor cells. For the colony formation assay, A549 (500 cells) and SPCA1 (500 cells) cells were seeded in 6-well plates and treated according to the experimental design. After 14 days, the proliferating cells were fixed with methanol for 30 min, followed by staining with 0.1% crystal violet for 2 h. The stained cells were photographed using a Zeiss LMS 700 (Carl Zeiss, Jena, Germany).

### 2.5. Cell Apoptosis Assay

The apoptotic ratios of A549 and SPCA1 cells were assessed after treatment with the indicated drugs using flow cytometry and propidium iodide (PI) apoptosis assays (Vazyme, Shanghai, China). For the flow cytometry assay, the logarithmic-growth cells were harvested and stained with PI and Annexin V-FITC for 15 min (Vazyme). FACS analysis was performed to detect the apoptotic ratio using an LSRII Flow Cytometer (BD, USA), and the data were analyzed using FlowJo software (Tree Star Inc., Ashland, OR, USA). For the PI apoptosis assay, 50% confluent cells were seeded in confocal plates and continuously cultured for 24 h. After being fixed with methanol for 30 min, cells were washed with PBST 3 times and then were incubated in PI solution for 15 min. Images of PI-positive cells were obtained using a Nikon Ti microscope (Nikon, Tokyo, Japan).

### 2.6. Xenograft Assay in Nude Mice

All animal experiments were approved by the Institutional Animal Care and Use Committee of Nanjing Medical University (IACUC: 2206030). The experimental BALB/c nude mice (male, 5 weeks old) were purchased from SLAC Laboratory Animal Center (Shanghai, China) and maintained in SPF facilities. To verify the effectiveness of NMN, 5 × 10^6^/100 μL of A549 cells and 5 × 10^6^/100 μL of SPCA1 cells were subcutaneously injected into the right and left axilla of the BALB/c nude mice, respectively. When the tumor volume (volume = length × width^2^/2) reached 100 mm^3^, the mice were randomly divided into four groups (*n* = 5) and were treated with solvent control, NMN (10 mM), NMN (100 mM), or NMN (100 mM) + Z-DEVD-FMK (50 mM) via intratumoral injection every 3 days. The body weight and tumor size were measured twice per week. After 4 weeks, the mice were humanely sacrificed, and the tumor weights were measured.

To further verify that NMN works through NAMPT’s catalysis, we first generated NAMPT^high^ A549 and SPCA1 cells using the NAMPT-specific plasmids via lentiviral infection. Subsequently, the A549 and NAMPT^high^ A549 cells were subcutaneously injected into the right and left axilla of BALB/c nude mice, respectively. The mice were randomly divided into three groups (*n* = 5) and were treated with solvent control, NMN (100 mM), or NMN (100 mM) + FK866 (30 nM) via intratumoral injection every 3 days. The SPCA1 and NAMPT^high^SPCA1 cells were treated analogously to the cells in the mice model.

### 2.7. Hematoxylin–Eosin (HE) Staining and Immunohistochemical (IHC) Assay

For the HE staining assay, xenograft tumors were sectioned and then stained with hematoxylin and eosin. For the IHC assay, the tumor sections were incubated with primary antibody ki67 (Cell Signaling Technology) and cleaved-caspase 3 (Cell Signaling Technology) overnight at 4 °C. After three successive rinses with PBS, the second antibody was added for 60 min at room temperature. The signal was amplified and visualized using 3′-diaminobenzidine chromogen (DAB).

### 2.8. Transmission Electron Microscopy (TEM)

A549 and SPCA1 cells were harvested after treatment with NMN for 48 h and then seeded on ACLAR film (Ted Pella, Redding, CA, USA). After 1 h, the cells were fixed in 2.5% glutaraldehyde in 0.1 M PBS at pH 7.4 for 2 h at 4 °C. Then, the specimens were embedded in Spurr’s resin and sectioned. The sections were stained with 2.5% uranyl acetate for 16 min, followed by counterstained lead citrate for 10 h. The intracellular structure was viewed using a TEM (JEOL JEM-ARM200F, Tokyo, Japan).

### 2.9. Cellular ROS Measurement

Cellular ROS levels were measured according to the manufacturer’s instructions (Beyotime). Briefly, A549 and SPCA1 cells were separately seeded with 50% confluent in 6-well plates and treated with 100 mM NMN for 48 h. Then, the cells were harvested and incubated with dichlorodihydrofluorescein diacetate (DCFH-DA) solution (Sigma-Aldrich) at 37 °C for 30 min. The cells were washed with PBS and subjected to flow cytometry to measure the levels of cellular ROS. In addition, the cells were seeded in confocal plates and incubated with DCFH-DA solution; then, they were imaged using a Nikon Ti microscope.

### 2.10. Lipid Peroxidation Assay

For the lipid peroxidation assay, A549 and SPCA1 cells were separately seeded with 50% confluent in confocal plates and treated with 100 mM NMN for 48 h. Then, the cells were incubated with non-oxidized C11-BODIPY 581/591 dye or oxidized C11-BODIPY 581/591 dye (MCE). After incubation at 37 °C for 30 min, the cells were imaged using a Nikon Ti microscope (Nikon, Tokyo, Japan).

### 2.11. Cellular Glutathione (GSH) and Malondialdehyde (MDA) Detection

NMN-treated A549 and SPCA1 cells were collected via centrifugation at 1000 rpm for 5 min at 4 °C. The cells were snap-frozen in liquid nitrogen and triturated. We added lysis buffer and then they were centrifuged at 12,000 rpm for 10 min at 4 °C. We removed the supernatant for later use. Then, cellular GSH and MDA were spectrophotometrically measured using a Glutathione Reductase Assay Kit (Beyotime) and a Lipid Peroxidation MDA Assay Kit (Beyotime).

### 2.12. Mito-FerroGreen-Labeled Immunofluorescence Assay (Fe^2+^)

A549 and SPCA1 cells were separately seeded on confocal plates and continuously cultured for 24 h. Then, cells were treated with separate drugs for another 48 h. Mito-FerroGreen solution (Dojindo, Japan) was placed at room temperature for at least 30 min and then added to the cells. The cells were immediately imaged using a Nikon Ti microscope.

### 2.13. ELISA Assay

The cellular NAM (Jingmei Biotechnology, Yancheng, Jiangsu, China), NMN (Jinmei Biotechnology), NR (Jinmei Biotechnology), NAD/NADH (Beyotime), and PUFA (Jinmei Biotechnology) were assayed using commercial ELISA kits according to the manufacturer’s instructions.

### 2.14. Statistical Analysis

All analyses were conducted and graphs were drawn using GraphPad Prism 8 and SPSS 20 software. Data are presented as the means ± standard deviation (SD). Student’s *t*-test was used to analyze the differences between the two groups, and one-way ANOVA was used for analyzing more than two groups. *p* < 0.05 was considered to be statistically significant. * *p* value < 0.05; ** *p* < 0.01; *** *p* < 0.001; **** *p* < 0.001; ns, no significance.

## 3. Results

### 3.1. High-Dosage NMN Inhibits Lung Cancer Growth through a Non-Apoptotic, Non-Autophagy, or Non-Necrosis Program

To identify the role of NMN in lung cancer cell growth under different dosing schemes, we conducted cell proliferation and inhibition experiments using various concentrations of NMN in A549 and SPCA1 cells, determined using a CCK8 assay. At low doses (10 and 20 mM) and prolonged exposure (48 h), NMN increased cell proliferation, but it induced the suppression of cell proliferation at the high dose (100 mM), as shown in Figure 1A,B. The finding was subsequently confirmed by the results of the tumor cell colony formation assay (Figure 1C). Moreover, high-dosage NMN enhanced cytotoxicity to lung adenocarcinoma cells compared with normal cells (Appendix A). Next, we performed PI stain assay in A549 and SPCA1 cells after NMN treatment and found that high-dose NMN could promote cell death. Moreover, high-dose NMN could further promote cell death after the inhibition of the apoptotic program via the apoptosis inhibitor Z-DEVD-FMK (Figure 1D–F). The findings suggest that high-dose NMN inhibits lung cancer cell growth through a non-apoptotic form of regulated cell death. To further test the hypothesis, we performed flow cytometry analysis and found that the apoptotic- or non-apoptotic-regulated death in cells after high-dose NMN treatment was higher than that in cells after solvent treatment; the number of non-apoptotic regulated death cells was much higher than the number of apoptotic cells after high-dose NMN treatment (Figure 1G–K). We also confirmed that NMN could promote cell death after inhibiting the apoptotic program only at the high dose (Figure 1G–K). The results of Western blot analysis showed that high-dose NMN had no notable drug-induced effects on apoptosis program molecules caspase3 and PARP (Figure 1L). In addition, PARPs are not exclusively involved in apoptosis induction but also in autophagic cell death and necrosis. These results also suggest that high-dose NMN has no noticeable effects on autophagic cell death and necrosis (Figure 1L).

Subsequently, we conducted in vivo xenograft experiments in mice by implanting A549 and SPCA1 cells to verify the effect of high-dose NMN on inhibiting lung cancer growth (Figure 2A). The results indicated that, with or without Z-DEVD-FMK treatment, NMN could inhibit tumor growth at high doses (Figure 2B–G). These data demonstrated that the effect of high-dose NMN in inhibiting lung cancer growth was independent of the apoptotic program. HE and ki67 staining of tumor samples showed that high-dose NMN treatment significantly increased intratumoral necrosis and suppressed tumor proliferation independent of the apoptotic program (Figure 2H,I). The results of caspase 3 staining for tumor samples further elucidated that NMN had no remarkable effects on the cell apoptosis program (Figure 2H,I). In summary, these data support that high-dosage NMN inhibits lung cancer growth through a non-apoptotic program.

### 3.2. High-Dosage NMN Inhibits Lung Cancer Growth by Inducing Ferroptosis Program

To clarify the effect of NMN in promoting cell death at high doses, apoptosis inhibitor Nec-1, autophagy inhibitor CQ, and ferroptosis inhibitor Fer-1 were administered separately with NMN to A549 and SPCA1 cells. The results of the CCK8 assay indicated that cotreatment with Nec-1 or CQ had no effect on the significant inhibition of cell growth by high-dose NMN, whereas Fer-1 protected the cells from high-dose NMN inhibition (Figure 3A,B). The results of the cell colony formation assay also confirmed that high-dose NMN failed to inhibit cell growth after blocking the ferroptosis program with Fer-1 (Figure 3C). In addition, NMN could accelerate A549 and SPCA1 cell death even with the treatment of ferroptosis inducers erastin and RSL3 (Appendix A). These results suggest that high-dose NMN induces ferroptosis in lung adenocarcinoma cells to inhibit cell growth.

Subsequently, we confirmed whether ferroptosis occurred after high-dose NMN treatment from the changes in cell morphology and composition. Using transmission electron microscopy (TEM), we observed cells showing typical characteristics of ferroptosis, with smaller mitochondria and increased membrane density, after high-dose NMN treatment (Figure 3D). We next detected cellular ROS levels using a DCFH-DA sensor through FACS and immunofluorescence analysis and found that total cellular ROS levels were significantly increased after high-dose NMN treatment (Figure 3E–G). We further evaluated the functional consequence of high-dose NMN treatment on lipid peroxide formation using a lipid peroxide sensor assay, in which oxidized C11-BODIPY sensors present green fluoresces and non-oxidized C11-BODIPY sensors present red fluoresces. As shown in Figure 3H, compared with the solvent treatment, high-dose NMN increased lipid peroxide accumulation in the A549 and SPCA1 cells. Moreover, NMN-treated A549 and SPCA1 cells were found to consume GSH and accumulate MDA (Figure 3I–L). A more intuitive finding was the accumulation of mitochondrial ferrous ions in A549 and SPCA1 cells after high-dose NMN treatment, which promoted ferroptosis (Figure 3M). These data support the notion that NMN inhibits lung adenocarcinoma growth by inducing the ferroptosis program.

### 3.3. High-Dosage NMN Inhibits Lung Cancer Growth through Its Metabolite NAM

Exogenous NMN is rapidly decomposed into NR and NAM in vivo; NR and NAM are converted into NMN through the catalysis of NRK or NAMPT after entering cells, and finally NAD^+^ is synthesized (Figure 4A). To clarify whether NMN or its metabolites NAM or NR play a crucial role in inhibiting tumor growth, we measured the contents of NAM, NR, and NMN in high-dose NMN-treated A549 and SPCA1 cells. We found that the metabolite NAM content significantly increased (Figure 4B); the contents of NMN and NR did not show notable changes (Figure 4C,D). In addition, as the final metabolite, the content of NAD^+^/NADH significantly decreased after high-dose NMN treatment (Figure 4E).

To further validate the putative hypothesis that NMN inhibits lung cancer growth through its metabolite NAM, we indirectly manipulated the amount of intracellular NAM by regulating the expression of catalyst NAMPT. As shown in Figure 4F, NAMPT^high^ A549 and SPCA1 cells were generated using NAMPT-specific plasmids. Then, an ELISA assay was conducted to detect the contents of intracellular NAM. We found that the conversion of NAM to NMN was accelerated after the overexpression of NAMPT, and NAM was eventually consumed. Conversely, when NAMPT expression was inhibited with the NAMPT inhibitor FK866, intracellular NAM was accumulated (Figure 4G,H). The results of the CCK8 and colony formation assays further showed that the growth rate of NAMPT^high^ A549 and SPCA1 cells was significantly higher than that of NAMPT^WT^ A549 and SPCA1 cells; however, when NAMPT inhibitor FK866 was added to NAMPT^high^ A549 and SPCA1 cells, the cell growth was again inhibited (Figure 4I–K). These results indicated that high-dose NMN inhibited lung cancer cell proliferation through its metabolite NAM.

Next, we validated the inhibitory effect of the NMN metabolite NAM on lung cancer growth in vivo. Xenograft experiments in mice were conducted by subcutaneously injecting NAMPT^WT^ or NAMPT^high^ A549 and SPCA1 cells in nude mice. As expected, we confirmed that the overexpression of NAMPT reversed the inhibition of tumor growth by NAM after high-dose NMN treatment, whereas NAM-restored cells combined with NAMPT inhibitor FK866 subsequently re-slowed the growth of tumors (Figure 5A–F). Intratumoral necrosis and tumor proliferation showed a consistent trend with tumor growth according to the results of HE and ki67 staining (Figure 5G,H). All of these data confirmed that high-dose NMN inhibited lung cancer growth through its metabolite NAM.

### 3.4. High-Dosage NMN Induces Ferroptosis via Its Metabolite NAM

To test the hypothesis that NAM plays a key role in suppressing lung cancer growth through the ferroptosis program, we analyzed cell composition changes by detecting cellular mitochondrial ferrous ions, ROS, MDA, PUFA, and GSH after over- or low-expression NAMPT with high-dose NMN treatment. The results showed that A549 and SPCA1 cells exhibited a reduction in cellular mitochondrial ferrous ion (Figure 6A), ROS (Figure 6B), MDA (Figure 6C), and PUFA (Figure 6D) contents and an accumulation of GSH (Figure 6E) after the overexpression of NAMPT accelerated the transformation of NAM, whereas the inhibition of NAMPT reversed the process (Figure 6A–E). The results of the 4-Hydroxynonenal (4-HNE) staining of the tumor samples showed that the overexpression of NAMPT reversed the induction of the ferroptosis program by NAM after high-dose NMN treatment, whereas NAM-restored cells, combined with the NAMPT inhibitor FK866, subsequently restarted the ferroptosis program (Figure 6F). These data show that NMN induces ferroptosis via its metabolite NAM at high concentrations.

### 3.5. High-Dosage NMN Promotes Ferroptosis through SIRT1–AMPK–ACC Signaling with NAM Overload

As the AMPK-mediated phosphorylation of ACC was reported to inhibit ferroptosis under energy stress [21], we investigated its role downstream of high-dose NMN action as a tumor suppressor. Consecutively, the tandem activation of SIRT1 and AMPK occurs in mammalian cells, where the increased effect can be inhibited with an overload of NAM [22]. These studies provided a theoretical basis for studying the anti-tumor effects of high-dose NMN as a function of ferroptosis-mediated lung adenocarcinoma cell death. As shown in Figure 7A, high-dose NMN inhibited the phosphorylation of AMPK and activated its downstream target, ACC, in vitro, in a dose-dependent manner. Then, under a high-dose NMN treatment, we used the SIRT1 agonist CAY10602, SIRT1 inhibitor selisistat, AMPK agonist GSK621, and AMPK inhibitor dorsomorphin to treat the lung adenocarcinoma cells, and evaluated the NAMPT, SIRT1, p-AMPK, AMPK, p-ACC, and ACC protein expressions using Western blot analysis. The results are shown in Figure 7B. CAY10602 promoted the expression of SIRT1 and the phosphorylation levels of AMPK, while suppressing the levels of p-ACC; selisistat treatment yielded the opposite expressions. In addition, during high-dose NMN treatment, GSK621 promoted the expression of p-AMPK and suppressed the expression of p-ACC, but showed no remarkable regulation of SIRT1; dorsomorphin reversed the regulation of p-AMPK and p-ACC by GSK621. The CAY10602, selisistat, GSK621, and dorsomorphin treatment did not affect the expression of NAMPT (Figure 7B). However, the overexpression of NAMPT activated the expression of p-AMPK and inhibited the expression of p-ACC in A549 and SPCA1 cells; NAMPT inhibitor FK866 produced the opposite result (Figure 7C). Neither CAY10602, Selisistat, GSK621, nor Dorsomorphin had any effect on the intracellular NAM content (Figure 7D,E). In summary, these results show that high-dose NMN treatment can activate SIRT1–AMPK–ACC signaling mediated through an overload of NAM.

Next, we explored the relationship between NMN-regulated NAM–SIRT1–AMPK–ACC signaling and ferroptosis. The results of the cell proliferation and apoptosis assays confirmed that the ferroptosis program promoted by SIRT1–AMPK–ACC signaling mediated by the overload of NAM was associated with lung adenocarcinoma cell proliferation and apoptosis in vitro (Figure 7F–H). We further analyzed cell composition changes by detecting cellular mitochondrial ferrous ions, ROS, lipid peroxidation, and PUFA after intervention with CAY10602, selisistat, GSK621, and dorsomorphinunder under high-dose NMN treatment in lung adenocarcinoma cells. The A549 and SPCA1 cells exhibited a reduction in cellular mitochondrial ferrous ions (Figure 8A), ROS (Figure 8B), lipid peroxidation (Figure 8C), and PUFA (Figure 8D,E) after CAY10602 or GSK621 treatment. Furthermore, the combined treatment of CAY10602 and dorsomorphin promoted cell ferroptosis, whereas the combined treatment of GSK621 and selisistat inhibited the ferroptotic program (Figure 8). These results further suggest that high-dose NMN treatment promotes ferroptosis through NAM-mediated SIRT1–AMPK–ACC signaling.

## 4. Discussion

The study findings showed that local treatment with high-dose NMN strongly inhibited tumor growth and induced tumor ferroptosis in lung adenocarcinoma cells. Specifically, we found that (i) high-dose NMN (100 mM) suppressed lung adenocarcinoma A549 and SPCA1 cell proliferation and promoted cell death, whereas low-dose (10 mM) treatment produced the opposite effect; (ii) the cell death induced by high-dose NMN occurred through ferroptosis; (iii) high-dose NMN treatment mainly induced intracellular NAM overload to trigger ferroptotic cell death; and (iiii) high-dose NMN induced tumor ferroptosis through an overload-NAM-mediated SIRT1–AMPK–ACC pathway. We further showed that the combination of NAMPT inhibitor FK866, a potential tumor therapy agent, with high-dose NMN local treatment increased the efficacy of tumor suppression in vitro and in vivo. Notably, NAMPT inhibitor FK866, as a monotherapy in clinical trials, failed to show notable anti-tumor action [23]. Thus, our findings suggest that lung adenocarcinoma patients should consider two points when using NMN: dose and compatibility with energy metabolism drugs.

Consistently, accumulating evidence demonstrates that NMN is often an anti-aging pharmaceutical associated with NAD^+^ biosynthesis to protect against damage and to promote cell proliferation through effectively elevating NAD^+^ levels under normal and pathophysiological conditions [5]. Tumors are critically associated with aging [12,24]. However, studies regarding NMN therapy for tumors are scarce. Pan et al. showed that NMN treatment (500 mg/kg/day) did not affect the growth of lung adenocarcinoma in mice via intraperitoneal injection [25], whereas Nacarelli T al. found that NMN (1 mM in vitro, 500 mg/kg/day in vivo by intraperitoneal injection) promoted tumor growth by governing the proinflammatory senescence-associated secretome [8]. Here, similar to its function in non-tumor cells, low-dose NMN promoted NAD^+^/NADH synthesis in A549 and SPCA1 cells (Figure 4E).

Most previous studies have shown that exogenous NMN cannot be directly used by cells and needs to be decomposed into NAM and NR, synthesized into endogenous NMN under the catalysis of NAMPT, and then converted into NAD^+^/NADH to exert an anti-aging effect and enhance cell proliferation [26,27]. However, NMN could be transported into cells through Slc12a8, whereas Slc12a8 knockdown abrogated the uptake of NMN in vitro and in vivo [28]. In our study, the results were in accordance with those of previous studies in that the short-term NMN administration most significantly increased NAM levels in A549 and SPCA1 cells. Notably, with high-dose NMN treatment, numerous sources of NAM could be made available to the cells for growth inhibition and death. As the catalysis proceeded, NAMPT maintained its population at a certain level as a rate-limiting step in NMN synthesis. This finding was confirmed by the results of immunoblotting experiments, where no significant difference was observed in NAMPT protein expression level after NMN treatment. This might have caused high-dose NMN-mediated self-damage in the A549 and SPCA1 cells due to excess NAM accumulation, which could not be metabolized by NAMPT.

NMN, a source of cellular energy, can regulate AMPK pathway proteins [29]. Here, NAM, one of its metabolites, intracellularly accumulated in massive amounts in lung adenocarcinoma cells and significantly inhibited SIRT1 activity. Additionally, a study demonstrated that AMPK phosphorylation was dependent on SIRT1 activity [30], which was verified in our study. Moreover, the AMPK-mediated phosphorylation of ACC facilitated polyunsaturated fatty acid synthesis and further promoted ferroptosis [21]. Again, a similarly direct correlation between NMN and ferroptosis primarily relies on an overload-NAM-mediated SIRT1–AMPK–ACC pathway. In addition, the ferroptosis phenotype was reversed by overexpressing NAMPT, which in turn resulted in a significant rescue of the proliferative capacity of lung adenocarcinoma cells, with the additional recovery of the tumor-promoting effects in vivo in the presence of high-dose NMN.

The caveat is that, unlike the systemic administration of NMN, we took advantage of local NMN delivery to investigate its metabolism and function in vitro and in vivo. Presently, the highest dose of NMN administered in various studies is 2000 mg/kg/day via oral gavage in rats, which contributes to reduced body weight gains, diminished food consumption, and increased ALT levels [31]. In one study, the authors showed few dose-limiting toxicities in NMN via intraperitoneal injection or oral administration. However, we found that exposure to high concentrations of NMN could destroy lung adenocarcinoma tissues through ferroptosis, with an increase in the destruction in tumor cells in comparison with the corresponding normal cells (Appendix A).

## 5. Conclusions

Overall, high-dose NMN treatment promoted ferroptosis to suppress lung adenocarcinoma growth through an excessive accumulation of NAM and the NAM-mediated SIRT1–AMPK–ACC pathway. The further exploration of a strategy to optimize the NMN response in patients with lung adenocarcinoma is warranted (Figure 9).

## Figures and Tables

**Figure 1 cancers-15-02427-f001:**
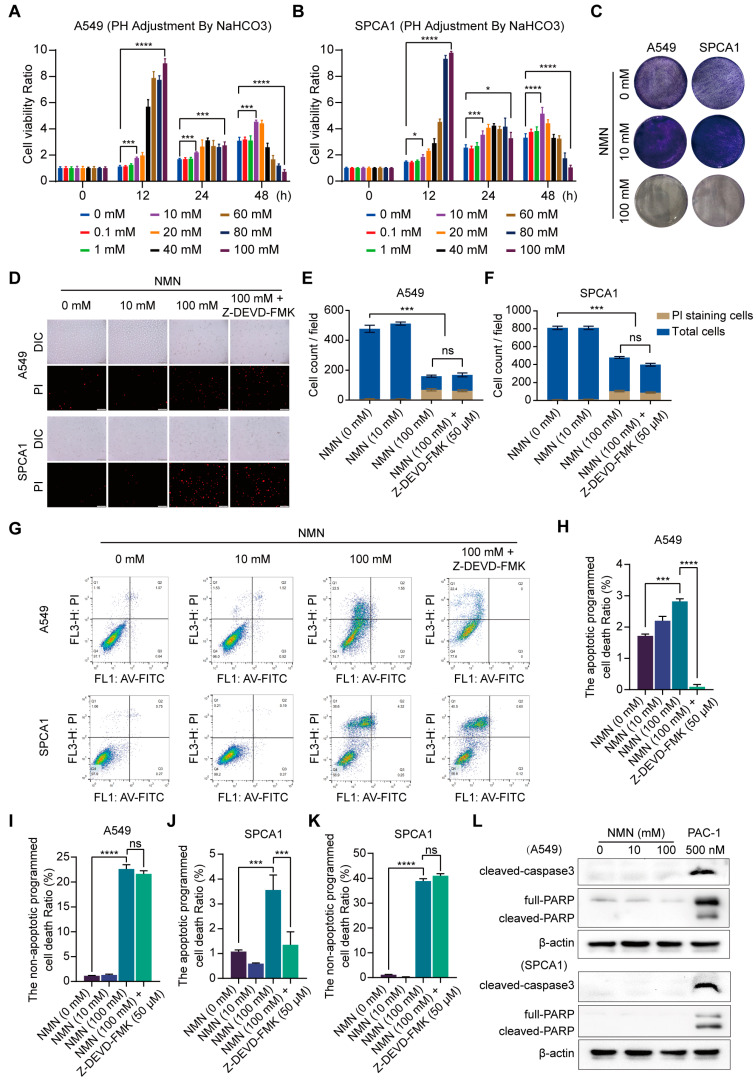
High-dose NMN induced non-apoptotic regulated death in lung adenocarcinoma cells. (**A**,**B**) Cell proliferation of A549 (**A**) and SPCA1 (**B**) cells was measured using CCK8 assay after treatment with different doses of NMN for indicated times (*n* = 5). (**C**) Representative images of colony formation for A549 and SPCA1 cells after different doses of NMN treatment. (**D**–**F**) Annexin V-FITC/PI double staining cytometry was analyzed for various NMN concentrations in presence or absence of caspase inhibitor (Z-DEVD-FMK). Representative fluorescence images of PI staining (**D**) and quantification of A549 (**E**) or SPCA1 (**F**) cell counts per field (*n* = 3); scale bar: 200 μm. (**G**–**K**) FACS of A549 and SPCA1 cells after different doses of NMN treatment in presence or absence of Z-DEVD-FMK. Representative images of FACS analysis (**G**). Quantification of apoptotic (**H**) or non-apoptotic (**I**) regulated death A549 cells and apoptotic (**J**) or non-apoptotic (**K**) regulated death SPCA1 cells (*n* = 3). (**L**) Western blot analysis of cleaved caspase-3 and PARP in A549 and SPCA1 cells. * *p* value < 0.05; *** *p* < 0.001; **** *p* < 0.0001; ns, no significance.

**Figure 2 cancers-15-02427-f002:**
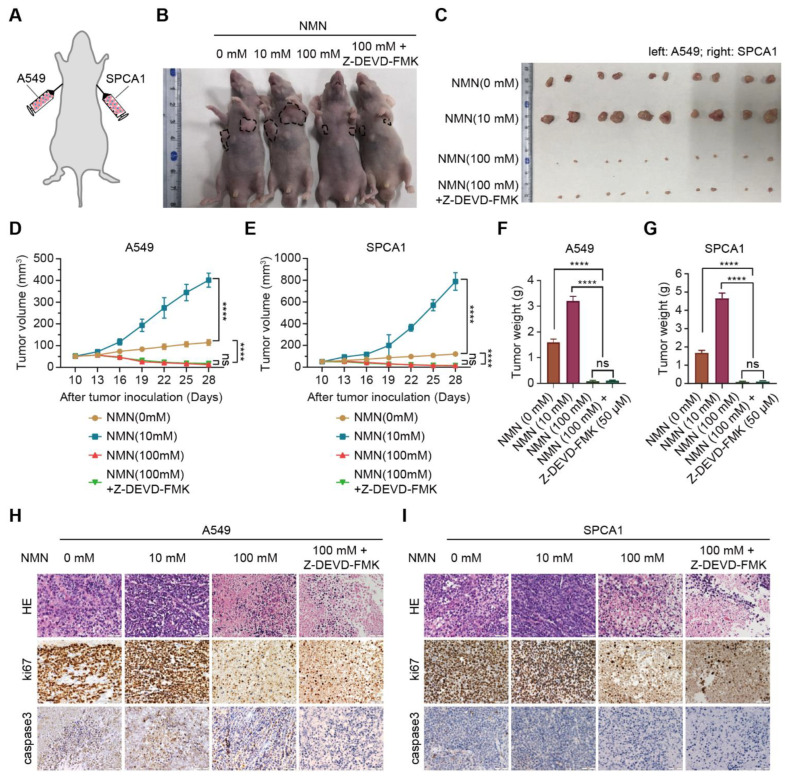
High-dose NMN inhibited lung adenocarcinoma growth in vivo. (**A**) Schematic representation of A549 and SPCA1 cells’ xenograft model. (**B**,**C**) Representative image of tumor-bearing BALB/C nude mice (**B**) and macroscopic image (**C**) of excised tumors in indicated groups. (**D**,**E**) Tumor growth curves of A549 (**D**) and SPCA1 (**E**) cells in indicated groups (*n* = 6). (**F**,**G**) Tumor weights of A549 (**F**) and SPCA1 (**G**) cells in indicated groups (*n* = 6). (**H**,**I**) Representative HE and immunohistochemistry staining of ki67 and caspase3 in A549 (**H**) and SPCA1 (**I**) tumor nodules; scale bar: 50 μm. **** *p* < 0.0001; ns, no significance.

**Figure 3 cancers-15-02427-f003:**
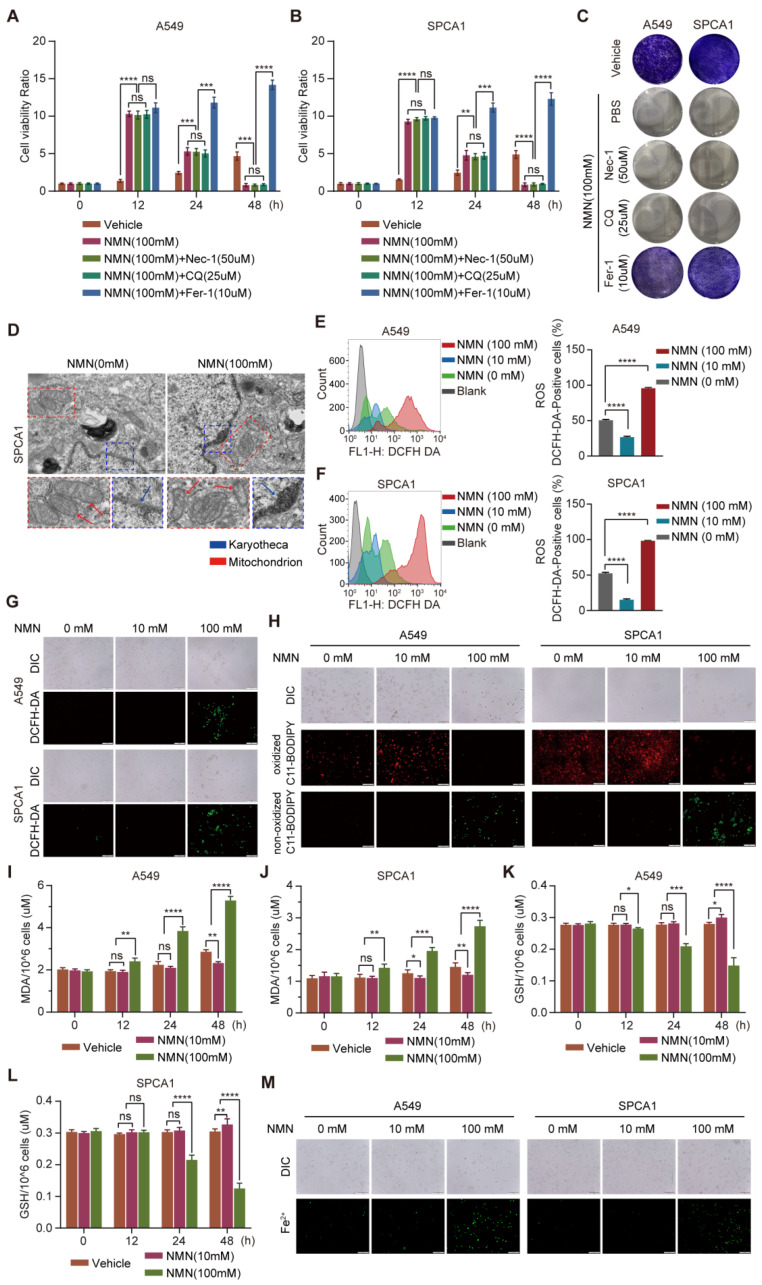
High-dose NMN induced ferroptosis of lung adenocarcinoma cells. (**A**,**B**) Cell proliferation of A549 (**A**) and SPCA1 (**B**) cells was measured using CCK8 assay after treatment with Nec-1, CQ, or Fer-1 combined with high-dose NMN for indicated durations (*n* = 5). (**C**) Representative images of colony formation for A549 and SPCA1 cells after treatment with Nec-1, CQ, or Fer-1 combined with high-dose NMN. (**D**) Representative images of mitochondrial and nuclear membrane morphology of SPCA1 cells via transmission electron microscopy analysis; scale bar: 200 nm. Blue arrows indicate karyotheca, and red arrows indicate mitochondria. (**E**–**G**) ROS production in A549 and SPCA1 cells induced by high-dose NMN was detected via DCFH-DA and measured via flow cytometry analysis (**E**,**F**) and immunofluorescence analysis (**G**); scale bar: 200 μm. (**H**) The effect of NMN on lipid peroxidation was detected in A549 and SPCA1 cells measured using BODIPY 581/591 C11 stain (BODIPY-C11); scale bar: 200 μm. (**I**,**J**) MDA levels in A549 (**I**) and SPCA1 (**J**) cells were examined using ELISA assay at different time points (*n* = 5). (**K**,**L**) GSH levels in A549 (**K**) and SPCA1 (**L**) cells were examined via ELISA assay at different time points (*n* = 5). (**M**) Mitochondrial iron concentration was determined via Mito-FerroGreen labeling using cell immunofluorescence; scale bar: 200 μm. * *p* value < 0.05; ** *p* value < 0.01; *** *p* < 0.001; **** *p* < 0.0001; ns, no significance.

**Figure 4 cancers-15-02427-f004:**
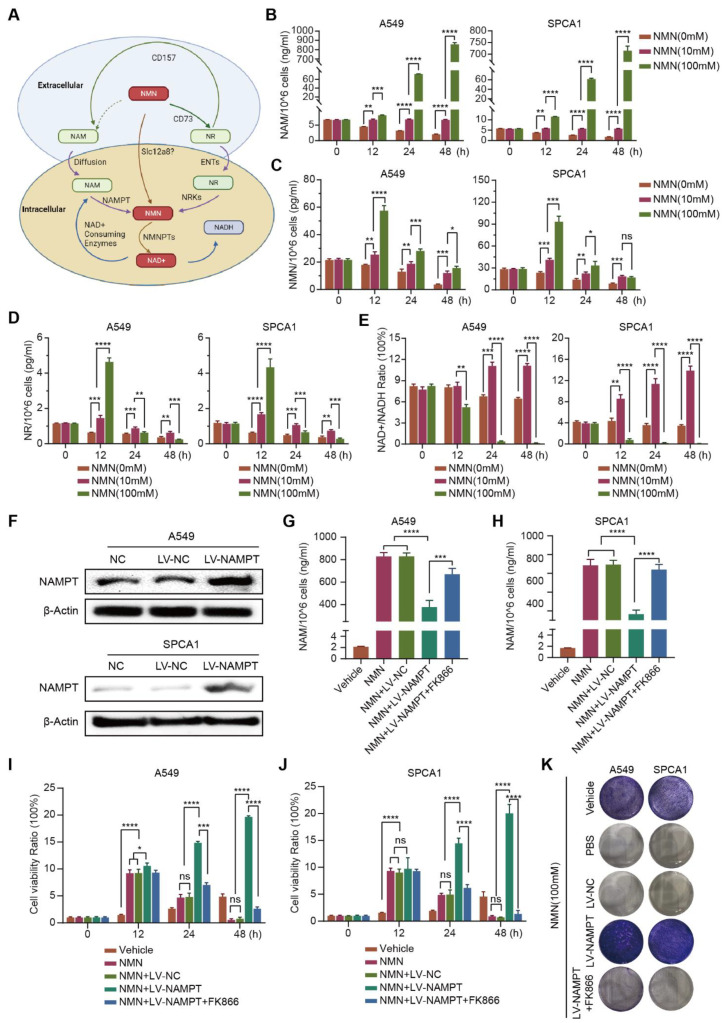
High-dosage NMN inhibits lung adenocarcinoma cell proliferation through its metabolite NAM. (**A**) Graphic illustration of intracellular NMN metabolism. (**B**–**E**) Concentrations of NMN metabolites NAM (**B**), NMN (**C**), and NR (**D**), along with the NAD^+^/NADH ratio (**E**) in A549 and SPCA1 cells after treatment with indicated concentrations of NMN, as measured using ELISA kits (*n* = 5). (**F**) Western blot analysis of NAMPT protein expression in A549 and SPCA1 cells after transfection with an overexpression of NAMPT or control lentivirus. (**G**,**H**) A549 (**G**) and SPCA1 (**H**) intracellular NAM contents were tested using ELISA kits in indicated groups (*n* = 5). (**I**,**J**) Cell proliferation of A549 (**I**) and SPCA1 (**J**) cells was measured using CCK8 assay in indicated groups (*n* = 5). (**K**) Representative images of colony formation for A549 and SPCA1 cells in indicated groups. * *p* value < 0.05; ** *p* value < 0.01; *** *p* < 0.001; **** *p* < 0.0001; ns, no significance.

**Figure 5 cancers-15-02427-f005:**
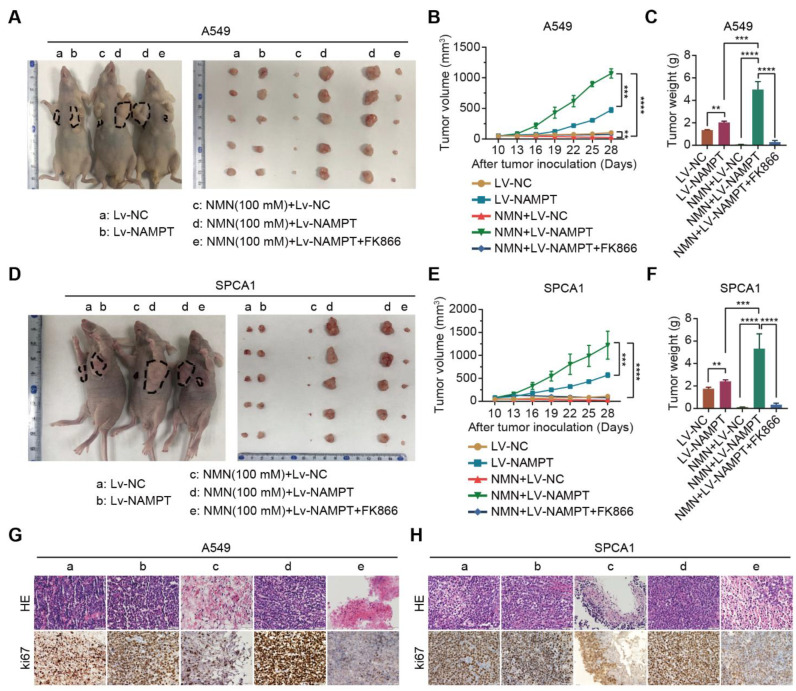
High-dosage NMN inhibits lung adenocarcinoma growth through its metabolite NAM in vivo. (**A**) Representative image of A549 xenograft model and macroscopic image of excised tumors in indicated groups. (**B**,**C**) Tumor growth curves (**B**) and tumor weight (**C**) in indicated groups (*n* = 6). (**D**) Representative image of SPCA1 xenograft model and macroscopic image of excised tumors in indicated groups. (**E**,**F**)**,** Tumor growth curves (**E**) and tumor weight (**F**) in indicated groups (*n* = 6). (**G**,**H**) Representative HE and immunohistochemistry staining of ki67 in the A549 (**G**) and SPCA1 (**H**) tumor nodules in indicated groups (a: Lv-NC, b: Lv-NAMPT, c: NMN (100 mM)+ Lv-NC, d: NMN (100 mM)+ Lv-NAMPT, e: NMN (100 mM)+ Lv-NAMPT+FK866); scale bar: 50 μm. ** *p* value < 0.01; *** *p* < 0.001; **** *p* < 0.0001.

**Figure 6 cancers-15-02427-f006:**
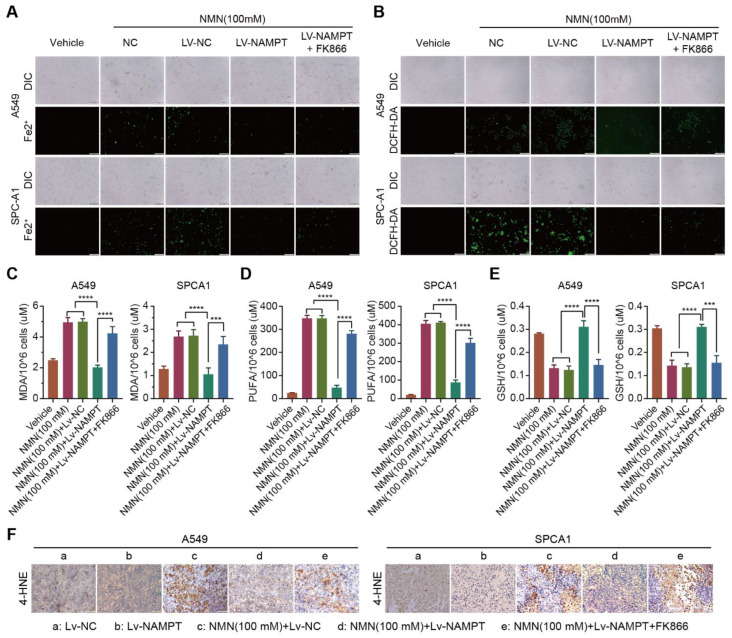
High-dosage NMN induces ferroptosis via its metabolite NAM. (**A**) The A549 and SPCA1 mitochondrial iron concentrations determined through Mito-FerroGreen labeling using cell immunofluorescence in indicated groups; scale bar: 200 μm. (**B**) ROS production in A549 and SPCA1 cells detected via DCFH-DA and immunofluorescence analysis in indicated groups; scale bar: 200 μm. (**C**–**E**) MDA (**C**), PUFA (**D**), and GSH (**E**) levels in A549 and SPCA1 cells examined via ELISA assay in indicated groups (*n* = 5). (**F**) Representative immunohistochemistry staining of 4-HNE in A549 and SPCA1 tumor nodules; scale bar: 50 μm. *** *p* < 0.001; **** *p* < 0.0001.

**Figure 7 cancers-15-02427-f007:**
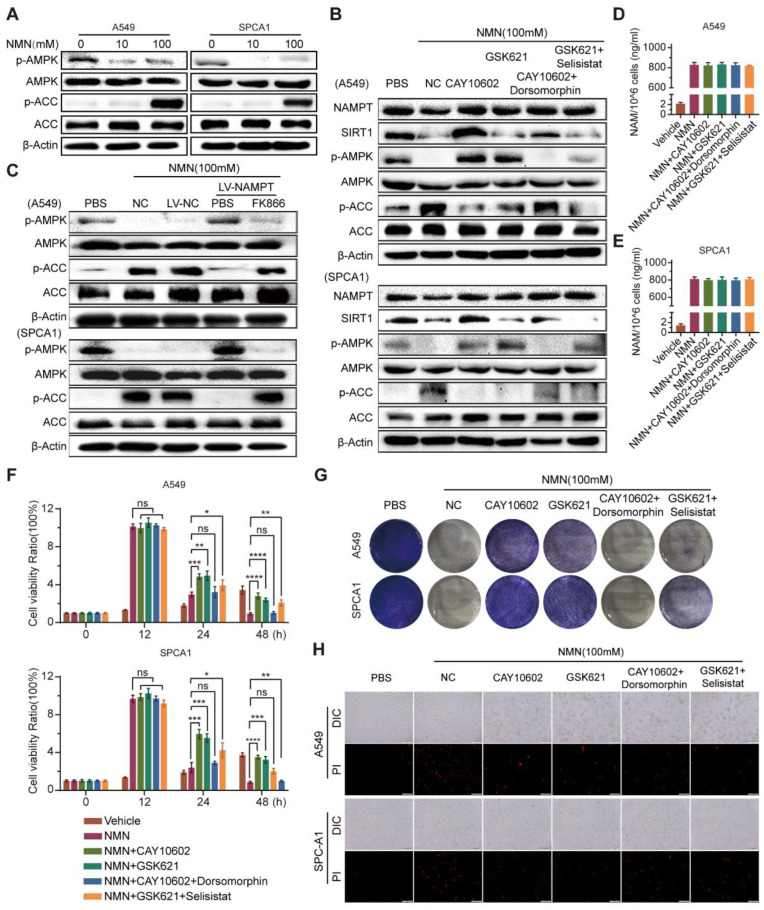
High-dosage NMN promoted ferroptosis of lung adenocarcinoma cells through NAM-overload-mediated SIRT1–AMPK–ACC pathway. (**A**) A549 and SPCA1 cells were treated with low- or high-dose NMN for 48 h; results of Western blot analysis of p-AMPK and p-ACC protein expression. (**B**) A549 and SPCA1 cells were treated with the SIRT1 agonist (CAY10602) or inhibitor (Selisistat), AMPK agonist (GSK621) or inhibitor (dorsomorphin), or combinations of the above agents, followed by high-dose NMN treatment for various intervals as indicated. Results of Western blot analysis of NAMPT, SIRT1, p-AMPK, and p-ACC protein expressions. (**C**) A549 and SPCA1 cells overexpressing NAMPT. The control was assessed with or without FK866 treatment, followed by high-dose NMN treatment for the indicated intervals. Results of Western blot analysis of p-AMPK and p-ACC protein expressions. (**D**,**E**) A549 (**D**) and SPCA1 (**E**) intracellular NAM contents were tested using ELISA kits in indicated groups (*n* = 5). (**F**) Cell proliferation of A549 and SPCA1 cells was measured using CCK8 assay in indicated groups (*n* = 5). (**G**) Representative images of colony formation for A549 and SPCA1 cells in indicated groups. (**H**) Annexin V-FITC/PI double staining cytometry was analyzed in indicated groups. Representative fluorescence images of PI staining; scale bar: 200 μm. * *p* value < 0.05; ** *p* value < 0.01; *** *p* < 0.001; **** *p* < 0.0001; ns, no significance.

**Figure 8 cancers-15-02427-f008:**
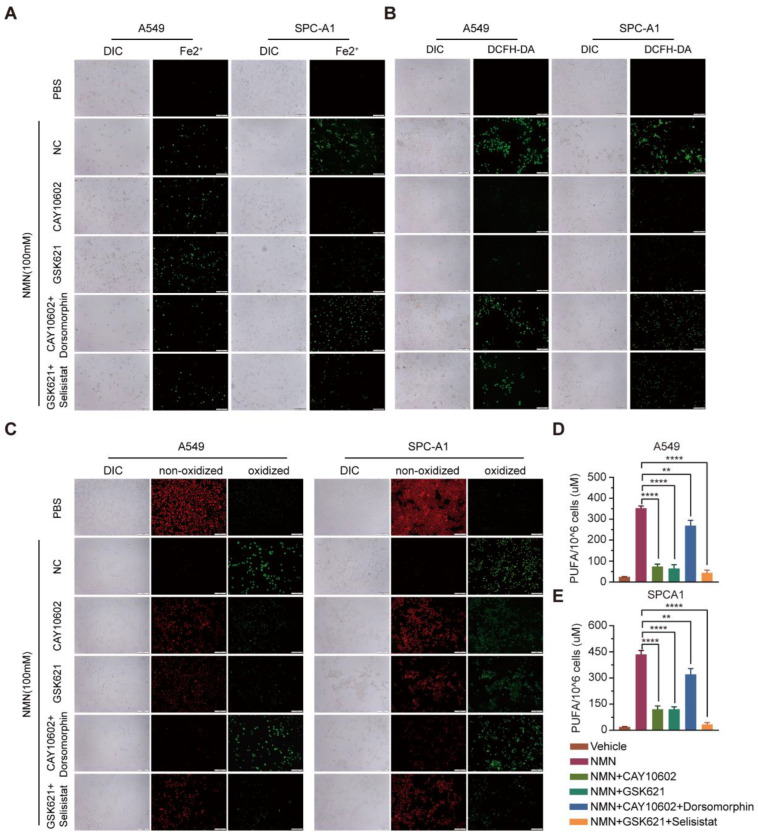
High-dosage NMN promoted ferroptosis of lung adenocarcinoma cells through NAM-overload-mediated SIRT1–AMPK–ACC pathway. (**A**) Mitochondrial iron concentration was determined via Mito-FerroGreen labeling using cell immunofluorescence in indicated groups; scale bar: 200 μm. (**B**) ROS production in A549 and SPCA1 cells detected via DCFH-DA, measured using immunofluorescence analysis in indicated groups; scale bar: 200 μm. (**C**) Lipid peroxidation was detected in A549 and SPCA1 cells measured using BODIPY 581/591 C11 stain (BODIPY-C11); scale bar: 200 μm. (**D**,**E**) PUFA levels in A549 (**D**) and SPCA1 (**E**) cells were examined using ELISA assay in indicated groups (*n* = 5). ** *p* value < 0.01; **** *p* < 0.0001.

**Figure 9 cancers-15-02427-f009:**
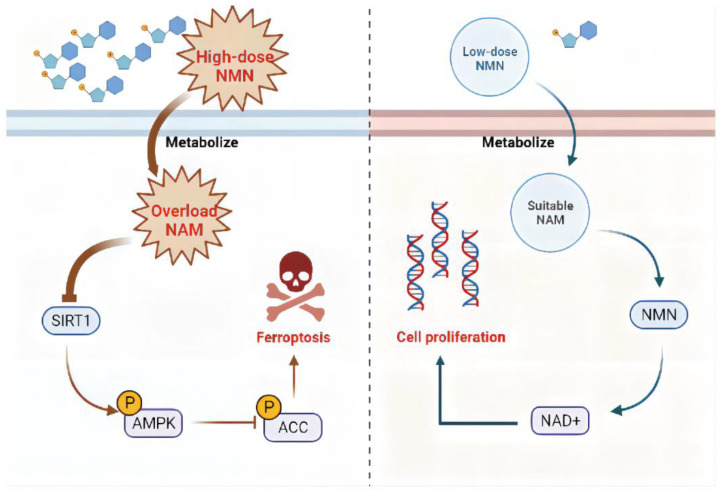
Proposed model for effects of NMN on regulating lung adenocarcinoma cell ferroptosis. In the context of high-dose NMN treatment (**left**), excess metabolized NAM inhibited AMPK phosphorylation by targeting SIRT1, triggering the process of ferroptosis via ACC activation pathways. In the context of low-dose NMN treatment (**right**), NAD^+^ was catalyzed in large quantities through intracellular NMN synthesized by a suitable concentration of NAM, leading to cancer cell proliferation.

## Data Availability

The raw data are available with the consent of the corresponding author.

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
