# Peer review of "High-Dosage NMN Promotes Ferroptosis to Suppress Lung Adenocarcinoma Growth through the NAM-Mediated SIRT1–AMPK–ACC Pathway"

_cancers, 2023, doi:10.3390/cancers15092427_

Round 1
Reviewer 1 Report
In this study, Zhang et al. demonstrated antitumor effects of NMN in lung adenocarcinoma in vitro and in vivo (mouse xenograft model). Antitumor effect of NMN was accomplished when applied in high dose. In addition, the authors have demonstrated that NMN induce ferroptosis through the SIRT1-3 AMPK-ACC pathway.
The study is well designed and results are interesting, however additional improve is required.
MINOR REVISIONS:
- English revision is necessary. In some cases, it is hard to understand the meaning, such as in: L69 (…accumulation of lipid-based…), L248, L300 (…a remarkable product of MDA…), L384-5, L452-4 etc.
- Many writing errors including unseparated words, unnecessary capitalization, sentences separations…
- L222: …the number of non-apoptotic programmed death cells… Change the term, i.e. to: … the ration of non-apoptotic dead cells…
- L298: …lipid peroxide ACCUMULATION…
- L73 – “peroxides” instead of “peroxidation”
- Missing references: L384, L54, L56, L74
- Citation number – in order of appearance (L107 – ref. 28)
- L461. Ref. 21 – it is cited in the text as ‘Timothy et al.” Please check as it does not correspond to the ref. 21 in the list.
- Please define abbreviations prior to first usage in the text or include in the abbreviations list: NR, STR, AMPK, SIRT1…
- According to the Nomenclature Committee on Cell Death (Galuzzi et al. 2012, 2018) a term programmed cell death should be used for physiological removal of cells during the development and normal cellular turnover only, while term regulated cell death (RCD) also includes forms of cell death triggered by intense intracellular or extracellular perturbations. Therefore, RCD is the more appropriate term to be used in this manuscript.
- Some micrographs presented in the figures should be larger since it is not possible or is hard to observe the described alterations: Fig.3D, 7H
- Fig. 7F – SPCA1 graph – overlapped columns at 48 h
- L434 lipid peroxidation is stated to be presented in Fig. 7L. Please check – Fig. 7K?
- L465 and 468 – no need for figures recalling in the Discussion section
- Materials and methods section:
o Please provide the data on town/state of companies
o L93: “the results were all negative, and the cells were free of mycoplasma contamination.” – unnecessary repeating, please correct
o L119 – please add the concentrations of the drugs
o Include immunohistochemistry method in the MM text. Also, please specify which anti-Caspase 3 antibody was used – against total or active form?
o L192 – Fe2+ assay is not immunocytochemistry assay. Please correct.
MAJOR REVISIONS:
- Introduction section (and Discussion, L458-9): It seems that too much attention is paid to the aging-related NAD+ deficiency and the relationship between aging and tumor development. Namely, the local treatment with high doses of NMN for lung carcinoma proposed in this study does not seem to involve the same mechanisms of action as the low-dose supplementation required for age-related conditions with NAM deficiency.
- L212: Authors claim that NMN when applied at low (10 mM) dose and prolonged exposure (72h) led to increased proliferation. However, at Fig. 1A and B only 0-48h time points are presented. Also, 10 mM is not the only dosage which provoked the increased cell proliferation. Please change in the text.
- L227: PARPs are not exclusively involved in apoptosis induction but also in autophagic cell death, necrosis etc. so please redefine the result.
- The results of the comparison of NMN effects on lung carcinoma and normal lung epithelial cells is missing (Suppl. Fig. 2). Please add in the Results sections (also in the MM section).
- Fig. 3D – micrographs are too small to observe the described ultrastructural alterations. Also, mitochondria of SPCA1 cells treated with 100 mM NMN does not look smaller when compared with control. Also, membrane density alteration seems not to be visible in the presented micrographs. If possible, quantify the results (i.e., average mitochondria surface or length).
- Fig. 7K – oxidized form of BODIPY in SPCA1 cells seems to be high in CAY10602 and GSK621 treatments and to overlap completely with the signal of non-oxidized form. Please check and change in the text if needed.
Reviewer 2 Report
The work explores the effect of high doses of NMN on two adenocarcinoma cell lines. It shows that 0.1 uM NMN inhibit cell growth by causing ferroptosis, while lower doses promote cell proliferation. The inhibitory effect is due to the nicotinamide (NAM) produced by the metabolism of NMN. NAM level is controlled also by the enzyme nicotinamide phosphoribosyl transferase (NAMPT). A NAM-mediated SIRT1-AMPK-ACC pathway promotes Ferroptosis.
- The authors should discuss how the 0.1 uM concentration of NMN they found to induce ferroptosis compares with the physio/pathological concentrations found in tissues, similarly for the NAM concentration.
- The switch from cancer-promoting to cancer-inhibiting cell proliferation of NMN is time and concentration-dependent thus is necessary to define the threshold of the switches before any suggestion of its potential use in adenocarcinoma therapy
- After the evidence that NAM is the active molecule, I am surprised that the authors did not treat the cells with various concentrations of NAM.
- The Mito-FerroGreen reagent used to monitor Fe(II) levels in the cells is to evaluate the mitochondrial iron level, which is not necessarily related to the cytosolic iron level. Thus, the data of figs 3M and 6A, B should be reevaluated. In addition, all the fluorescent figures seem too dark, and difficult to evaluate both in print and on the screen.
- There are a number of non-explained acronyms and abbreviations that do not facilitate the reading, for example, NAM, NR, ACC, STR analysis, FL1-H
- At Line 272: «prolonged exposure (72 h)» but the figures show data only up to 48 h.
- Fig 1A,B «PH Adjustment by NaHCO3)» explain in the text.
Round 2
Reviewer 1 Report
Although the authors resolved many suggestions/issues, some are still needed to be resolved prior to the publication.
- English should be revised by an expert, some sentences are hard to understand.
- USA instead of America (M&M section)
- programmed cell death - replace with regulated cell death through the text
- Many abbreviations are still missing from the body text.
- Abstract background - No direct relationship between NMN and cancer was provided in the abstract background.
- Results subtitle "High-Dosage NMN inhibits lung cancer growth through a non-apoptotic program" should be revised since PARP is included. In addition, the PARP role and conclusion is not properly incorporated in the text.
- Fig.3D - no difference between NAM-treated and untreated cells is visible from the micrographs. Please revise (also in the text). Also, the authors claim that increased membrane density is detectable at TEM level in the treated cells, however, that is not showed in the micrographs.
Reviewer 2 Report
The answers to the points I have raised are not satisfactory and there was only little effort to improve the manuscript.
- Point 1 was to know which is the basal concentration of NMN in cells and tissues to be compared with the concentration that is added in the experiments. The intracellular NMN concentration after the treatment expressed in ng/mL is not relevant.
- To answer point 2 the authors added the sentence “Thus, our study suggests that lung adenocarcinoma patients should consider two points when using NMN: dose and compatibility with energy metabolism drugs.” but I do not understand what it means. What are the two points? The proposal to treat adenocarcinoma with a chemical that favor cell proliferation does not seem very wise!
- I expected an explanation for not using NAM in the system, but I do not think NAMPT is a substitute for it.
- In the legend of fig. 8 is still stated “Intracellular iron concentration was determined by Mito-FerroGreen labeling using cell immunofluorescence in indicated groups” which is wrong since Mito-FerroGreen detects mitochondrial iron. I do not see much improvement in the figures
- In the simple summary NAM, NAMPT and others should be open. "NAM-medicated" is for NAM-mediated, I guess.
- In the rebuttal, you stated that corrections are in yellow, but they are in red.
Round 3
Reviewer 2 Report
the text was modified as suggested and improved